

# Mass Accommodation and Gas-Particle Partitioning in Secondary Organic Aerosols: Dependence on Diffusivity, Volatility, Particle-phase Reactions, and Penetration Depth

Manabu Shiraiwa[1,*] and Ulrich Pöschl[2,*]

1. Department of Chemistry, University of California, Irvine, CA92625, USA

2. Multiphase Chemistry Department, Max Planck Institute for Chemistry, 55128 Mainz, Germany

* Correspondence to: m.shiraiwa@uci.edu; u.poschl@mpic.de



**Abstract.**
Mass accommodation is an essential process for gas-particle partitioning of organic compounds in
secondary organic aerosols (SOA). The mass accommodation coefficient is commonly described
as the probability of a gas molecule colliding with the surface to enter the particle phase. It is often
applied, however, without specifying if and how deep a molecule has to penetrate beneath the
surface to be regarded as incorporated into the condensed phase (adsorption vs. absorption). While
this aspect is usually not critical for liquid particles with rapid surface-bulk exchange, it can be
important for viscous semisolid or glassy solid particles to distinguish and resolve the kinetics of
accommodation at the surface, transfer across the gas-particle interface, and further transport into
the particle bulk.
For this purpose, we introduce a novel parameter: an effective mass accommodation coefficient
$\alpha_{\text{eff}}$ that depends on penetration depth and is a function of surface accommodation coefficient,
volatility, bulk diffusivity, and particle-phase reaction rate coefficient. Application of $\alpha_{\text{eff}}$ in the
traditional Fuchs-Sutugin approximation of mass-transport kinetics at the gas-particle interface
yields SOA partitioning results that are consistent with a detailed kinetic multilayer model (KM-
GAP, Shiraiwa et al. 2012) and two-film model solutions (MOSAIC, Zaveri et al., 2014) but
deviate substantially from earlier modeling approaches not considering the influence of penetration
depth and related parameters.
For highly viscous or semisolid particles, we show that the effective mass accommodation
coefficient remains similar to the surface accommodation coefficient in case of low-volatile
compounds, whereas it can decrease by several orders of magnitude in case of semi-volatile
compounds. Such effects can explain apparent inconsistencies between earlier studies deriving
mass accommodation coefficients from experimental data or from molecular dynamics
simulations.
Our findings challenge the approach of traditional SOA models using the Fuchs-Sutugin
approximation of mass transfer kinetics with a fixed mass accommodation coefficient regardless
of particle phase state and penetration depth. The effective mass accommodation coefficient
introduced in this study provides an efficient new way of accounting for the influence of volatility,
diffusivity, and particle-phase reactions on SOA partitioning in process models as well as in
regional and global air quality models.



## Introduction.

Secondary organic aerosols (SOA) are major constituents of atmospheric particulate matter, affecting air quality, climate, and public health (Jimenez et al., 2009; Kanakidou et al., 2005; Pöschl and Shiraiwa, 2015; Shrivastava et al., 2017a). Gas-phase reactions of volatile organic compounds (VOC) emitted from various anthropogenic and biogenic sources with oxidants such as ozone and OH radicals lead to the formation and growth of SOA (Kroll and Seinfeld, 2008). The oxidation of VOC forms a myriad of semi-volatile (SVOC) and low volatility organic compounds (LVOC) that can condense on pre-existing particles (Ziemann and Atkinson, 2012) or contribute to nucleation and new particle formation (Tröstl et al., 2016). The evolution of SOA is a complex multi-step process that involves chemical reactions and mass trnsport in the gas phase, at the particle surface and in the particle bulk, but the interplay of these processes and the rate-limiting steps in SOA formation have not yet been fully resolved/elucidated (Shiraiwa et al., 2014).

Traditionally, SOA particles were assumed to be homogeneous and well-mixed quasi-liquid droplets (Pankow, 1994). As demonstrated by recent atmospheric measurements and laboratory experiments, they can adopt glassy solid or amorphous semi-solid phase states, challenging the traditional views of SOA properties, interactions and effects (Koop et al., 2011; Reid et al., 2018; Virtanen et al., 2010). Slow diffusion of water, oxidants and organic molecules in viscous, semi-solid, or glassy particles may lead to kinetic limitations in heterogeneous and multiphase reactions (Alpert et al., 2019; Davies and Wilson, 2015; Kuwata and Martin, 2012; Shiraiwa et al., 2011; Zhang et al., 2018; Zhou et al., 2019). Global model calculations suggest that the phase state of atmospheric SOA may vary between liquid, semi-solid and solid in the planetary boundary layer, while SOA should be mostly in a glassy state in the free troposphere (Shiraiwa et al., 2017). The occurrence of glassy SOA in the free troposphere may promote ice nucleation and cloud droplet activation (Knopf et al., 2018; Slade et al., 2017) and facilitate long-range transport of toxic organic compounds contained in SOA (Mu et al., 2018; Shrivastava et al., 2017b).

The formation and properties of SOA are large sources of uncertainty in the current understanding of global air quality, climate change, and public health. The development of SOA models is among the most challenging problems in atmospheric chemistry (Tsigaridis et al., 2014). In most current air quality, atmospheric chemistry and climate models, the limiting step of SOA



formation is assumed to be gas-phase oxidation of VOC to form semi-volatile and low volatile
products. Thus, gas-phase oxidation is described kinetically, while gas-particle partitioning is often
approximated by quasi-instantaneous equilibrium partitioning of the oxidation products (Pankow,
1994; Shrivastava et al., 2017a; Tsigaridis et al., 2014). The assumption of quasi-instantaneous
gas-particle equilibration, however, is in question if particles are highly viscous, semi-solid or
glassy - especially at low temperatures and low relative humidity (RH) (Li and Shiraiwa, 2019;
Shiraiwa and Seinfeld, 2012). Experimental studies found kinetic limitations for gas uptake and
particle evaporation at low RH (Liu et al., 2016; Perraud et al., 2012; Vaden et al., 2011; Yli-Juuti
et al., 2017), but not for mixing in SOA at medium or high RH (Ye et al., 2016; Ye et al., 2018).
An appropriate treatment of kinetic limitations depending on ambient conditions is critical for
accurately reproducing particle size distribution dynamics in SOA growth (Shiraiwa et al., 2013a;
Zaveri et al., 2018; Zaveri et al., 2020).
The dynamics of gas-particle partitioning have been considered in a wide range of
atmospheric aerosol models, including aerosol dynamics models (Liu et al., 2019; McVay et al.,
2014; Pandis et al., 1993; Riipinen et al., 2011; Zaveri et al., 2014), kinetic multilayer models
(Berkemeier et al., 2016; Fowler et al., 2018; Roldin et al., 2014; Shiraiwa et al., 2012), GECKO-
A (Aumont et al., 2005), the volatility basis set approach (Trump and Donahue, 2014; Trump et
al., 2014), the statistical oxidation model (Cappa et al., 2016; Jathar et al., 2016), and particle
evaporation models (Vaden et al., 2011; Yli-Juuti et al., 2017).  Most model studies are using the
Fuchs-Sutugin approximation of mass-transport kinetics at the gas-particle interface with a fixed
mass accommodation coefficient that does not vary with particle phase state nor with the volatility
and diffusivity of the investigated organic compounds. Molecular dynamics simulations (Julin et
al., 2014; Von Domaros et al., 2020) and a recent SOA chamber study (Liu et al., 2019) suggest
that the mass accommodation coefficients for semi-volatile organic molecules on organic
substrates are close to unity. Measurement-derived mass accommodation coefficients reported
from thermodenuder investigations of SOA volatility distributions, however, were one to three
orders of magnitude lower (Kostenidou et al., 2018; Lee et al., 2010; Saleh et al., 2011).
Overall, the relations between particle phase state, mass accommodation, and the growth
and atmospheric evolution of SOA have not yet been resolved and continue to be a subject of
scientific debate. In this study, we investigate the influence of volatility, diffusivity, and particle
phase state on the mass accommodation and gas-particle partitioning of organic compounds in





SOA by detailed and simplified kinetic modeling approaches, comparing the Fuchs-Sutugin
approximation to a detailed kinetic multilayer model (KM-GAP, Shiraiwa et al. 2012) as well as
approximate and transient two-film model solutions (MOSAIC, Zaveri et al., 2014).

**Theory and Methods**
Traditionally, dynamic models of aerosol chemistry and physics describe the rate of gas-
particle partitioning by a first-order gas-particle mass transfer coefficient ($k_{gp}$ in s$^{-1}$) based on the
Fuchs-Sutugin approximation of gas diffusion in the transition regime (Seinfeld and Pandis, 2016):
$$k_{gp} = 4\,\pi\,D_g\,r_p\,N_p\,\beta \qquad (1)$$
$$\beta = \frac{0.75\,\alpha\,(1+Kn)}{Kn^2 + Kn + 0.283\,Kn\,\alpha + 0.75\alpha} \qquad (2)$$
where $D_g$ (cm$^2$ s$^{-1}$) is the gas phase diffusivity, $r_p$ (cm) is the particle radius, $N_p$ (cm$^{-3}$) is the particle
number concentration, $Kn$ is the Knudsen number, and $\alpha$ is the mass accommodation coefficient
which represents the probability for a gas molecule colliding with the surface of the particles to
enter the condensed phase. $Kn$ is the ratio of the mean free path in the gas phase ($\lambda$), which can be
calculated using the mean thermal velocity ($\omega$), and the particle radius: $Kn = \lambda\,/\,r_p = 3\,D_g\,/\,(\omega\,r_p)$
(Pöschl et al., 2007). Note that $k_{gp}$ is also often termed as condensation sink (CS). According to
the absorptive partitioning theory under the assumption of ideal mixing (Pankow, 1994; Trump
and Donahue, 2014), the rate of change of the gas- and particle-phase mass concentrations ($C^g$, $C^p$)
of an organic compound in SOA partitioning can be expressed as:
$$\frac{dC^g}{dt} = -k_{gp}\left(C^g - \frac{C^p}{C_{OA}}C^0\right) \qquad (3)$$
$$\frac{dC^p}{dt} = k_{gp}\left(C^g - \frac{C^p}{C_{OA}}C^0\right) - k_b C^p \qquad (4)$$
where $C_{OA}$ ($\mu$g m$^{-3}$) is the organic aerosol particle mass concentration, $C^0$ ($\mu$g m$^{-3}$) is the gas phase
saturation mass concentration of the pure organic compound, and $k_b$ (s$^{-1}$) is the first-order rate
coefficient for its reaction in the particle bulk. The term $\frac{C^p}{C_{OA}}C^0$ represents gas-phase concentration
of Z right at the surface and condensation is driven by gas-phase concentration gradient of Z
between the gas and condensed phases.
While the term mass accommodation coefficient is widely used in atmospheric aerosol
studies, its precise meaning is not always well defined. In particular, $\alpha$ is often applied without
specifying if and how deep a molecule has to penetrate beneath the surface to have entered the




condensed phase (adsorption vs. absorption). This aspect is usually not critical for liquid droplets
with rapid surface-bulk exchange, fast bulk diffusion, and swift equilibration between the
condensed phase and the surrounding gas phase. For viscous or solid particles, however, it can be
essential to distinguish and resolve the kinetics of surface and bulk processes, including
accommodation at the surface, transfer across the gas-particle interface, and further transport into
the particle bulk (Kolb et al., 2010; Pöschl et al., 2007; Shiraiwa et al., 2012).
Building on the PRA kinetic model framework (Pöschl et al., 2007) and the kinetic
multilayer model of surface chemistry and gas-particle interactions in aerosols and clouds (KM-
GAP; Shiraiwa et al., 2012), we have derived an expression for the mass accommodation
coefficient as a function of penetration depth into the particle bulk and related parameters (see
step-by-step derivation in Appendix):
$$\alpha(x) = \alpha_s \frac{1}{1 + \frac{\alpha_s\, \omega\, C^0}{4\, D_b\, \rho_p}\, x \cdot 10^{-12}\, \frac{\mathrm{g\ cm^{-3}}}{\mathrm{\mu g\ m^{-3}}}} \qquad (5)$$

Here $\omega$ (cm s$^{-1}$) is the mean thermal velocity of the organic compound in the gas phase, $D_b$ (cm$^2$
s$^{-1}$) is its diffusivity in the condensed phase, $\rho_p$ (g cm$^{-3}$) is the particle density, and $x$ (cm) is the
penetration depth. The scaling factor $10^{-12}$ (g cm$^{-3}$)/($\mu$g m$^{-3}$) allows for inserting $C^0$ in the
commonly used units of $\mu$g m$^{-3}$; it can be omitted when $C^0$ is inserted in g cm$^{-3}$ or when all
quantities are inserted with standard SI units (cgs or mks system of units).
The surface accommodation coefficient $\alpha_s$, which corresponds to $\alpha(0)$ with the penetration
depth of 0, is the probability for a gas molecule Z colliding with the surface not to be immediately
scattered back to the gas phase but to be accommodated at the surface for period longer than the
duration of an elastic scattering process (Pöschl et al., 2007). Various equivalent, similar or closely
related terms and parameters have been defined and used in the scientific literature include (Kolb
et al., 2010; Pöschl et al., 2007): the condensation coefficient (Pruppacher and Klett, 1997),
adsorption coefficient (Shi et al., 1999; Turco et al., 1989; Worsnop et al., 2002), sticking
coefficient (Hanson, 1997), sticking probability (Clement et al., 1996; Garrett et al., 2006),
trapping probability (Masel, 1996), adsorptive mass accommodation coefficient (Elliott et al.,
1991), and thermal accommodation coefficient (Li et al., 2001; Worsnop et al., 2002).
When the penetration depth equals one or two molecular layers, i.e., once or twice the
effective molecular length or diameter ($\delta$), the corresponding penetration-depth-dependent mass


accommodation coefficient is equivalent to the quasi-static surface accommodation coefficient
($\alpha_{ss}$) or bulk accommodation coefficient ($\alpha_b$), respectively, as defined in earlier kinetic multilayer
model studies (Shiraiwa et al., 2012): $\alpha(\delta) = \alpha_{ss}$ and $\alpha(2\delta) = \alpha_b$. A recent study has compared this
kinetic multilayer (KM) modeling approach with molecular dynamics (MD) simulations to
calculate mass accommodation coefficients for a variety of semi-volatile compounds with different
volatilities in squalene (Von Domaros et al., 2020). The penetration depth was assumed to be equal
to the sum of half of the molecule's own length and half of the length of a squalene molecule. For
the evaluation of uncertainties and sensitivities, the penetration depth was also varied from semi-
volatile molecule's own length as a lower bound to the half-width of the nonuniform free energy
region determined by the MD free energy profile as an upper bound. Within this range, the results
of MD and KM simulations were in good agreement with each other, confirming the consistency
and validity of the multilayer approach (Von Domaros et al., 2020).

Using the two-film theory of mass transfer between gas and particle phase, Zaveri et al.

(2014) showed that the effects of a concentration gradient in the particle can be represented by a
thin film adjacent to the surface with the following thickness or effective penetration depth for
non-reactive partitioning and reactive uptake, respectively:
$$x_{eff} = r_p / 5 \qquad \text{(non-reactive partitioning)} \qquad (6)$$
$$x_{eff} = r_p \left( \frac{1 - Q}{q \coth q - 1} \right) \quad \text{(reactive uptake)} \qquad (7)$$
where $Q$ is the ratio of the average particle-phase concentration to the surface concentration at
steady state and $q$   is a dimensionless diffusion-reaction parameter (Seinfeld and Pandis, 2016):
$$Q = 3 \left( \frac{q \coth q - 1}{q^2} \right) \qquad (8)$$
$$q = r_p \sqrt{\frac{k_b}{D_b}} \qquad (9)$$
Note that $q_Z$ is the ratio of the particle radius to the so-called reacto-diffusive length, $(D_b/k_b)^{0.5}$,
representing the characteristic depth to which a species can penetrate while reacting in the particle
bulk (Pöschl et al., 2007; Worsnop et al., 2002).

By inserting $x_{eff}$ in equation (5), we obtain an effective mass accommodation coefficient

that accounts for the influence of penetration depth and its dependence on the diffusivity and
reactivity of the investigated chemical species in the particle:


$$\alpha_{\text{eff}} = \alpha\,(x_{\text{eff}}) \tag{10}$$

**Results and Discussion**
To investigate and demonstrate the relevance of the kinetics of mass accommodation and
the applicability of $\alpha_{\text{eff}}$, we simulate the temporal evolution of partitioning and equilibration of
semi-volatile organic compounds (SVOC) with $C^0 = 100$ µg m$^{-3}$ and $D_g = 0.1$ cm$^2$ s$^{-1}$ interacting
with non-volatile seed particles with a number concentration of 5000 cm$^{-3}$, an initial diameter of
200 nm, and a surface accommodation coefficient $\alpha_s = \alpha(0) = 1$. For the SVOC, we assume initial
gas- and particle-phase concentrations of 2 µg m$^{-3}$ and 0 µg m$^{-3}$, respectively. The particles are
assumed to be either liquid with a bulk diffusion coefficient $D_b = 10^{-7}$ cm$^2$ s$^{-1}$ or semisolid with $D_b$
$= 10^{-15}$ cm$^2$ s$^{-1}$.
Model calculations were performed with the detailed kinetic multilayer model of gas-
particle interactions (KM-GAP, Shiraiwa et al. 2012), with the Model for Simulating Aerosol
Interactions and Chemistry (MOSAIC; Zaveri et al., 2014), and with an aerosol dynamic model
using the simple Fuchs-Sutugin gas-phase diffusion model (F-S) with different values of $\alpha_m$. Here,
the KM-GAP results can be regarded as a benchmark, because the KM-GAP model explicitly
resolves all relevant processes - including gas diffusion, reversible adsorption, surface-bulk
exchange, bulk diffusion, and condensed-phase reactions - and has been successfully validated by
against experimental data of both non-reactive partitioning and reactive gas uptake in a wide range
of aerosol and surrogate systems (Berkemeier et al., 2017; Shiraiwa et al., 2012). The MOSAIC
model yields approximate and transient solutions building on a less detailed representation of gas-
particle interactions, which does not resolve reversible adsorption and surface-bulk exchange
(Zaveri et al., 2014). In the F-S approximation, the kinetics of particle-phase mass transport are
represented only by $\alpha_m$ as inserted into Eq. (2).
For liquid particles with fast surface-bulk exchange and bulk diffusion ($D_b = 10^{-7}$ cm$^2$ s$^{-1}$),
$\alpha(x)$ remains close to $\alpha_s = \alpha(0) = 1$, and all models yield the same result of fast mass transfer from
the gas to particle phase and equilibration within one second. For semi-solid particles with $D_b =$
$10^{-15}$ cm$^2$ s$^{-1}$, however, the temporal evolution of the SVOC gas-phase and particle-phase
concentrations varies between different models and different values of $\alpha$ as shown in Figure 1 on
logarithmic scales.





According to KM-GAP (black line), the initial uptake of SVOC by the semisolid particle
phase is as fast as approximated by F-S with $\alpha = \alpha_{ss} = \alpha_Z(\delta) = 3\times10^{-2}$ corresponding to a
penetration depth of only one molecular length (monolayer) below the particle surface. After one
second, however, the KM-GAP uptake is limited by bulk diffusion and slows down substantially.
After about one hour, KM-GAP converges with the F-S approximation using $\alpha = \alpha_{eff} = \alpha(r_p/5) =$
$8\times10^{-4}$. Notably, the F-S approximation with $\alpha_{eff}$ is identical to the MOSAIC approximation,
although the latter is based on different rate equations using a unity mass accommodation
coefficient like KM-GAP ($\alpha_s = 1$) and a two-film approach of bulk diffusion (Zaveri et al., 2014).
The MOSAIC transient solution exhibits a very high and likely overestimated initial uptake
corresponding to the F-S approximation with $\alpha = \alpha_s = 1$, because it does not resolve reversible
adsorption and desorption at the surface (Shiraiwa et al., 2012). After ~1 min, however, the
MOSAIC transient solution converges with KM-GAP. Overall, Figure 1 demonstrates that
accurate modeling of SVOC partitioning and uptake into semisolid particles requires an explicit
treatment of reversible adsorption and desorption at short time scales (< 1 min) and an explicit
treatment of bulk diffusion at intermediate time scales (~1 min to ~1 h). At long timescales (> 1
h), the partitioning is reasonable well captured by both the MOSAIC approximation using a two-
film approach of bulk diffusion (Zaveri et al., 2014) as well as the simple F-S approximation
accounting for the influence of penetration depth with the effective mass accommodation
coefficient, $\alpha_{eff}$, newly introduced this study.
Figure 2a shows the temporal evolution of the gas-phase concentration of organic
compounds with different volatilities ($C^0 = 0.1$ to 1000 µg m$^{-3}$) that undergo non-reactive
partitioning into semisolid seed aerosol particles ($D_b = 10^{-15}$ cm$^2$ s$^{-1}$). At short timescales,
substantial deviations can occur for semi-volatile compounds ($C^0 = 1$ to 100 µg m$^{-3}$), but at longer
time scales KM-GAP and the F-S approximation with $\alpha_{eff}$ are in reasonably good agreement
(relative deviations <10% after ~1 h). For low-volatile compounds ($C^0 < 1$ µg m$^{-3}$), equilibration
is achieved faster than for semi-volatile compounds because local thermodynamic equilibrium
between the gas phase and the particle surface is quickly established by condensation without
significant re-evaporation (Li and Shiraiwa, 2019; Zaveri et al., 2014). Semi-volatile compounds
with reactive functional groups can undergo particle-phase reactions such as dimerization and
oligomerization (Ziemann and Atkinson, 2012). Peroxide-containing highly oxidized molecules


(HOM) are labile with chemical half-lives shorter than one hour (Krapf et al., 2016; Tong et al.,
2016; Tong et al., 2019), and a recent study has shown that particle-phase reactions must be
considered to describe HOM effects on particle growth (Pospisilova et al., 2020). Model results
for SVOC partitioning plus reactive uptake with different rate coefficients in semisolid aerosol
particles are shown in Figure 2b. The results of the Fuchs-Sutugin approximation with $\alpha_{eff} = \alpha(x_{eff})$
and $x_{eff}$ from Eq. 7 are identical to the MOSAIC approximate and transient solutions. The uptake
predicted by KM-GAP is similar but slightly slower in case of high bulk reaction rate coefficients,
which can be attributed to the influence of reversible adsorption and desorption at the surface.
Additional simulations with $\alpha_s = 0.1$ confirm that the results of the Fuchs-Sutugin approximation
with $\alpha_{eff}$ and the MOSAIC approximate solution are identical, and that the results of KM-GAP and
the MOSAIC transient solution are similar (Fig. S1).
For a given surface accommodation coefficient of $\alpha_s = 1$, which is likely a good
approximation for SVOC on organic surfaces (Julin et al., 2014; Von Domaros et al., 2020),
Figures 3a and 3b show how the effective mass accommodation coefficient $\alpha_{eff}$ depends on
volatility and bulk diffusivity as related to particle phase state and viscosity according to the
Stokes-Einstein relation (Shiraiwa et al., 2011). In the liquid phase with high bulk diffusivity ($D_b$
$> 10^{-10}$ cm$^2$ s$^{-1}$), $\alpha_{eff}$ is essentially the same as $\alpha_s$ independent of volatility ($\alpha_{eff} \approx \alpha_s \approx 1$). With a
decrease of bulk diffusivity in viscous or semisolid particles, $\alpha_{eff}$ decreases substantially for SVOC
($0.3 < C^0 < 300$ µg m$^{-3}$) and so-called intermediate volatility organic compounds (IVOC; $300 < C^0$
$< 3 \times 10^6$ µg m$^{-3}$) but not for LVOC ($3 \times 10^{-4} < C^0 < 0.3$ µg m$^{-3}$) and so-called extremely low-volatile
organic compounds (ELVOC; $C^0 < 3 \times 10^{-4}$ µg m$^{-3}$). The reason why compounds with higher
volatility are more strongly affected by particle phase state and diffusivity is that they are more
likely to desorb back to the gas phase when diffusion into the bulk is slow. Compounds with lower
volatility exhibit much lower desorption rates and are less likely to re-evaporate even if their
diffusion into the bulk is slow. On the other hand, the influence of particle phase state and
diffusivity increases with particle size because longer pathways of diffusion are required for
effective accommodation, penetration, and absorption of gas molecules into larger particles as
illustrated in Figures 3c and 3d.
The theoretically predicted influence of volatility on effective mass accommodation is
consistent with a recent experimental study of α-pinene SOA reporting that the observed mass
accommodation coefficients decreased from ~1 for low-volatile compounds to ~0.3 for semi-



volatile compounds (Liu et al., 2019). Particle viscosity and bulk diffusivity were not reported for
these experiments, but values around $10^{-13}$ to $10^{-14}$ cm$^2$ s$^{-1}$ had previously been estimated for the
diffusion coefficient of organic compounds in α-pinene SOA under dry conditions (Zhou et al.,
2013). As illustrated in Figure 4a, theoretical predictions of $\alpha_{\text{eff}}$ assuming $\alpha_s = 1$ and $D_b = 10^{-12}$ to
$10^{-14}$ cm$^2$ s$^{-1}$ approximately capture the decrease and encompass the variability and uncertainty
range of the experimentally derived mass accommodation coefficients reported by (Liu et al.,
2019). Indeed, the observational $\alpha_m$ values reported in (Liu et al., 2019) and other experimental
studies are usually obtained by fitting measurement data with the F-S approximation, and thus they
should be directly compared to effective mass accommodation coefficient $\alpha_{\text{eff}}$ as derived by
integration of the F-S approximation with detailed kinetic models of mass transport across the gas-
particle interface. Figure 4b shows a wide range of other measurement-derived mass
accommodation coefficients for various SOA and surrogate systems (data points/shaded areas) in
comparison to generic values of $\alpha_{\text{eff}}$ (lines) calculated for characteristic experimental conditions
($\omega = 2.0 \times 10^4$ cm s$^{-1}$, $\rho_p = 1$ g cm$^{-3}$, and $r_p = 100$ nm, and $D_b = 10^{-19}$ to $10^{-5}$ cm$^2$ s$^{-1}$). As indicated
by molecular dynamics simulations and related studies, the surface accommodation coefficient
(adsorption probability) for semi-volatile or low-volatile organic compounds on organic surfaces
is likely close to unity, $\alpha_s = 1$ (Julin et al., 2014; Von Domaros et al., 2020). Accordingly, low
observational values of $\alpha$ can be attributed to the penetration-depth dependence of mass
accommodation and plausibly explained by different scenarios/combinations/ratios of volatility
and diffusivity, which can lead to a substantial decrease of $\alpha_{\text{eff}}$ relative to $\alpha_s$ in semi-solid particles.
With regard to the dependence of $\alpha_{\text{eff}}$ on $C^0$, mixing effects and non-ideality may lead to deviations
between $C^0$ and $C^*$ (Zuend and Seinfeld, 2012), which should be taken into account in further
investigations of mass accommodation and its influence on the formation and growth of SOA
particles.

On the other hand, high reactivity can compensate the influence of low diffusivity and mass

transport limitations in the particle phase, keeping $\alpha_{\text{eff}}$ close to $\alpha_s$. In case of non-reactive
partitioning, the effective penetration depth used to calculate $\alpha_{\text{eff}}$ is one fifth of the particle radius,
i.e., $x_{\text{eff}}/r_p = 0.2$ (Eq. 6). In case of reactive uptake, however, $x_{\text{eff}}$ decreases with increasing
reactivity and with decreasing diffusivity according to Eqs. (7) to (9). Figure 5a illustrates how the
effective penetration depth normalized by particle radius, $x_{\text{eff}}/r_p$, decreases with increasing first-



order bulk reaction rate coefficient, $k_b$, and with decreasing diffusion coefficient, $D_b$. The reduced
effective penetration depths at high $k_b$ and low $D_b$ reflect that reactive uptake by semisolid particles
proceeds mainly through chemical reaction near the surface (Shiraiwa et al., 2013a). Figure 5b
illustrates how $\alpha_{eff}$ depends on volatility and diffusivity for reactive uptake with $\alpha_s = 1$ and a first-
order bulk reaction rate coefficient $k_b = 0.1$ s$^{-1}$. In comparison to Fig. 3b for non-reactive
partitioning, Fig. 5b shows that particle phase reactivity leads to an extension of the volatility-
diffusivity parameter space where $\alpha_{eff} \approx \alpha_s$ (red area): For semi-solid particles with low diffusivity,
the parameter range of strong deviations between $\alpha_{eff}$ and $\alpha_s$ (yellow/green/blue area) is shifted
towards higher volatility (lower right corner).

**Summary and conclusions**

Traditional SOA modeling approaches are often using the Fuchs-Sutugin approximation of

mass-transport kinetics at the gas-particle interface in combination with mass accommodation
coefficients that are not appropriately defined, leading to inconsistent results and conclusions. To
overcome such deficiencies and difficulties, we have introduced an effective mass accommodation
coefficient $\alpha_{eff}$ that depends on penetration depth and is a function of surface accommodation
coefficient, volatility, bulk diffusivity, and particle-phase reaction rate coefficient. Application of
$\alpha_{eff}$ in the traditional F-S approximation of SOA partitioning yields results that are consistent with
detailed kinetic multilayer models (KM-GAP; Shiraiwa et al., 2012) and two-film models
(MOSAIC, Zaveri et al., 2014)).

We suggest that $\alpha_{eff}$ and its dependence on penetration depth and related parameters should

be applied and considered when the F-S approximation is used to investigate and simulate gas-
particle interactions in viscous or semi-solid organic aerosols. The simple parameterization can be
incorporated into regional and global models for a more realistic representation of SOA processes
in the atmosphere, which seems particularly important with regard to the ubiquity of amorphous
semi-solid or glassy particles predicted for the free troposphere as well as planetary boundary layer
air at low relative humidity and low temperature (Maclean et al., 2017; Shiraiwa et al., 2017).

In the analysis and interpretation of SOA chamber and laboratory experiments, $\alpha_{eff}$

provides a simply way of accounting for the potential impact of volatility, diffusivity, and particle
phase state on the kinetics of gas-particle partitioning for analysis and interpretation of chamber
experiments. In particular, it may help to address and resolve apparent inconsistencies between the



definitions and parameter values of mass accommodation coefficients that are derived from
experimental data and from molecular dynamics simulations.

At short timescales, however, $\alpha_{eff}$ is not sufficient to properly describe the kinetics of gas-

particle interactions with the F-S approximation. Such conditions require detailed kinetic model
simulations with kinetic multilayer models or equivalent approaches explicitly resolving mass
transport at the surface and in the bulk of the particle. The same applies for particles with layered
structures such as surface crusts (solid/viscous surface layers) that may form upon chemical aging
and can strongly impact the uptake of semi-volatile compounds and multiphase chemical processes
in the particle phase (Pfrang et al., 2011; Vander Wall et al., 2018; Zhou et al., 2019). Moreover,
mixed organic-inorganic particles often undergo liquid-liquid phase separation at moderate and
high RH (Krieger et al., 2012; You et al., 2014; Zuend and Seinfeld, 2012), and liquid-liquid phase
separation can also occur for purely organic particles (Song et al., 2017). The interplay of particle
phase state and phase separation can further impact SOA partitioning (Shiraiwa et al., 2013b). In
such complex particle morphologies with multiple phases, gradients and discontinuities of
diffusivity may occur within the particle bulk and require more advanced modeling approaches of
gas-particle interaction kinetics to be addressed in future studies.



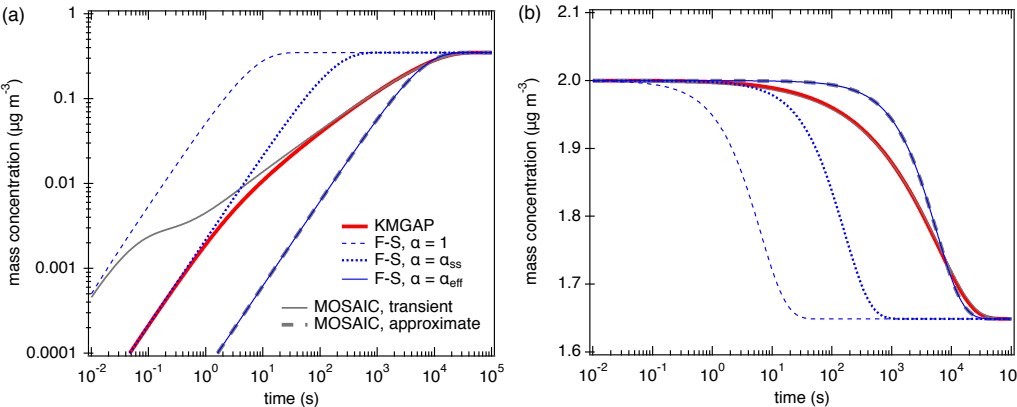

**Figure 1.** Temporal evolution of the particle phase concentration (a) and the gas phase concentration (b) of semi-volatile organic compounds (SVOC, $C^0 = 100$ μg m$^{-3}$) interacting with semisolid seed aerosol particles ($D_b = 10^{-15}$ cm$^2$ s$^{-1}$, $\omega = 2\times10^4$ cm s$^{-1}$, $\rho_p = 1$ g cm$^{-3}$). The red lines are simulation results of KM-GAP, and the blue lines are the results of an aerosol dynamic model that employs the Fuchs-Sutugin approximation with $\alpha = \alpha_s = 1$ (dashed), $\alpha = \alpha_{ss} = 3\times10^{-2}$ (dotted), and $\alpha = \alpha_{eff} = 8\times10^{-4}$ (solid). The gray lines represent the MOSAIC transient solution (solid) and approximate solution (dashed) (Zaveri et al., 2014).





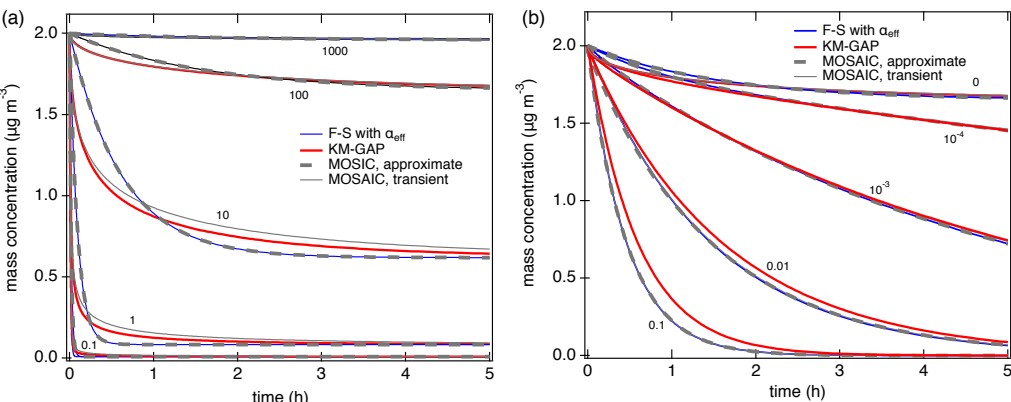

**Figure 2.** Temporal evolution of the gas phase concentration of organic compounds interacting with semisolid seed aerosol particles ($\alpha_s = 1$, $\omega = 2\times10^4$ cm s$^{-1}$, $D_b = 10^{-15}$ cm$^2$ s$^{-1}$, $\rho_p = 1$ g cm$^{-3}$): (a) Non-reactive partitioning of compounds with different volatilities ($C^0 = 0.1$, 1, 10, 100, 1000 µg m$^{-3}$); (b) reactive uptake of semi-volatile compounds ($C^0 = 100$ µg m$^{-3}$) with different first-order bulk reaction rate coefficients ($k_b = 0$, $10^{-4}$, $10^{-3}$, 0.01, 0.1 s$^{-1}$). The red lines are simulation results of KM-GAP, and the blue lines are the results of an aerosol dynamic model that employs the Fuchs-Sutugin approximation with $\alpha_{eff} = \alpha(r_p/5)$ for non-reactive partitioning (a) and with $\alpha_{eff} = \alpha(x_{eff})$ and $x_{eff}$ from Eq. (5) for reactive uptake (b). The gray lines represent the MOSAIC transient solution (solid) and approximate solution (dashed) (Zaveri et al., 2014).



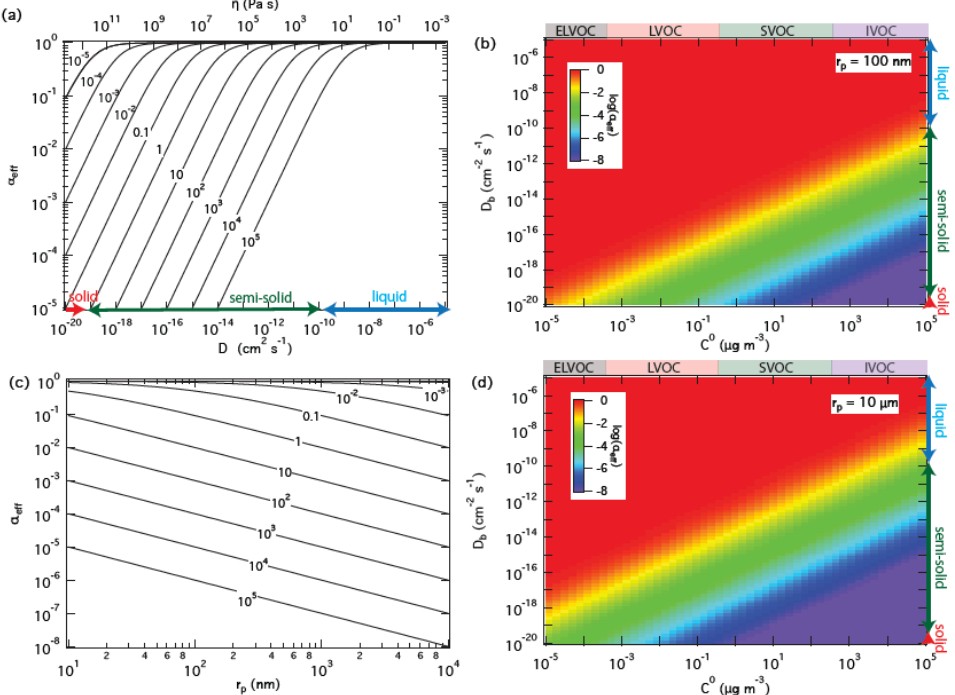


**Figure 3.** Effective mass accommodation coefficients, $\alpha_{eff}$, for non-reactive partitioning of organic
compounds Z ($\alpha_s = 1$, $\omega = 2 \times 10^4$ cm s$^{-1}$) with liquid, semi-solid, or solid aerosol particles ($\rho_p = 1$
g cm$^{-3}$) depending on pure compound volatility, $C^0$, particle bulk diffusivity, $D_b$ (corresponding to
viscosity, $\eta$), and particle radius, $r_p$: $\alpha_{eff}$ calculated as a function of $D_b$ for $C^0 = 10^{-5}$ to $10^5$ µg m$^{-3}$
with $r_p = 100$ nm (a); $\alpha_{eff}$ calculated as a function of $C^0$ and $D_b$ with $r_p = 100$ nm (b)  and 10 µm
(d); $\alpha_{eff}$ calculated as a function of particle radius for $D_b = 10^{-15}$ cm$^2$ s$^{-1}$ and different levels of
volatility ($C^0 = 10^{-3}$ to $10^5$ µg m$^{-3}$) (c).



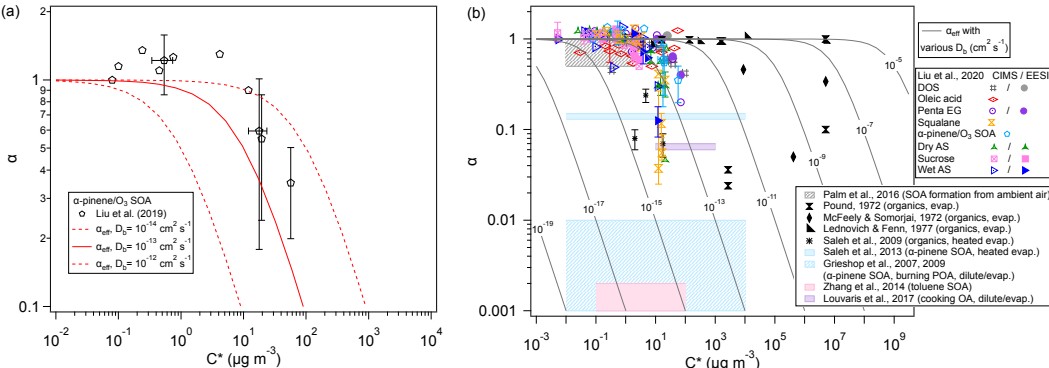

**Figure 4.** Effective mass accommodation coefficients, $\alpha_{eff}$ (lines, Eqs. 5-10) compared to measurement-derived mass accommodation coefficients, $\alpha$ (data points/shaded areas, Eqs. 1-2), plotted against effective saturation mass concentration, $C^*$, for various SOA and surrogate systems assuming $\alpha_s = 1$, $\omega = 2\times10^4$ cm s$^{-1}$, $\rho = 1$ g cm$^{-3}$, $r_p = 100$ nm, and $C^0 = C^*$: (a) observational results from laboratory experiments with semi-volatile components of SOA generated by ozonolysis of $\alpha$-pinene (data points, (Liu et al., 2019)) compared to $\alpha_{eff}$ for $D_b = 10^{-14}$ to $10^{-12}$ cm$^2$ s$^{-1}$ (lines); (b) observational results from earlier experimental investigations of laboratory-generated and ambient samples (data points/shaded areas, compiled by Liu et al., 2019) compared to generic values of $\alpha_{eff}$ for $D_b = 10^{-19}$ to $10^{-5}$ cm$^2$ s$^{-1}$ (lines).






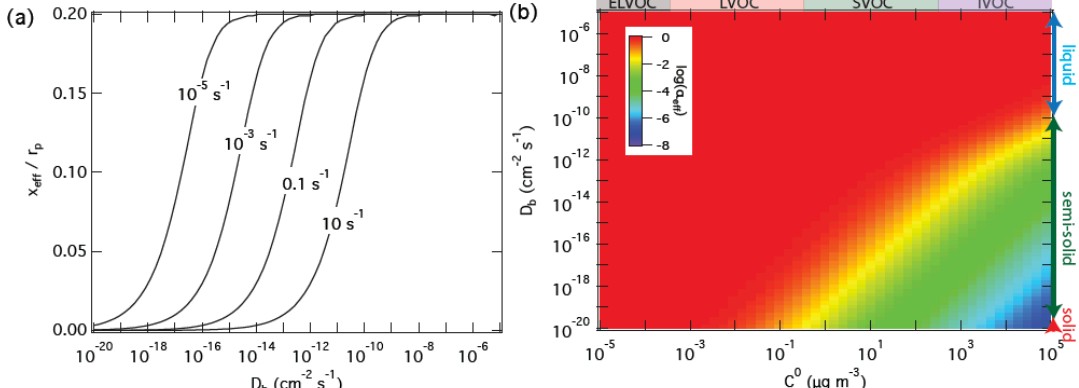



**Figure 5.** Effective penetration depths normalized by particle radius, $x_{eff}$, and mass
accommodation coefficients, $\alpha_{eff}$, for reactive uptake of organic compounds $Z$ ($\alpha_s = 1$, $\omega = 2 \times 10^4$
cm s$^{-1}$) by liquid, semi-solid, or solid aerosol particles ($r_p = 100$ nm, $\rho_p = 1$ g cm$^{-3}$) depending on
pure compound volatility, $C^0$, particle bulk diffusivity $D_b$, and first-order bulk reaction rate
coefficient, $k_b$: (a) $x_{eff}$ calculated as a function of $D_b$ and $k_b = 10^{-5}$ to $10$ s$^{-1}$; (b) $\alpha_{eff}$ calculated as a
function of $C^0$ and $D_b$ for $k_b = 0.1$ s$^{-1}$.






**Appendix:**

**Derivation of penetration-depth-dependent mass accommodation coefficient**

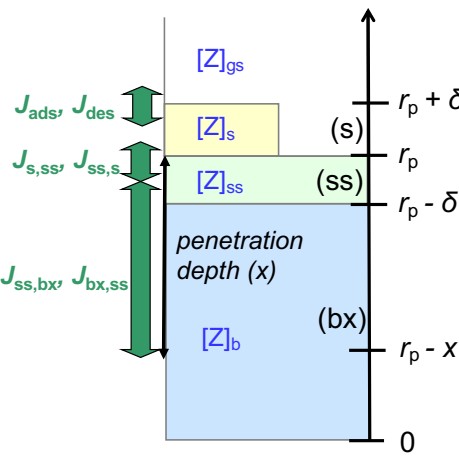

**Figure A1.** Schematic illustration of the kinetic multilayer modelling approach resolving mass transport fluxes ($J$) between the near-surface gas phase (gs), the sorption layer (s), the quasi-static surface layer (ss), and the bulk layer at penetration depth $x$ (bx) (Shiraiwa et al., 2012).

Figure A1 illustrates the applied kinetic multi-layer model framework, in which the structure and composition of a particle are described by a sorption layer (s), a quasi-static surface layer (ss), multiple bulk layers (b), and any volatile, semi-volatile, or low-volatile chemical species (Z) that can undergo gas-particle partitioning and transport between the different layers and chemical reactions with each other (Pöschl et al., 2007; Shiraiwa et al., 2012). At low gas-phase concentration levels or high surface bulk exchange rates (e.g., for liquid particles under dilute atmospheric conditions), surface coverage and saturation effects can be neglected, and the surface accommodation coefficient ($\alpha_\mathrm{s}$) approaches the parameter value for an adsorbate-free surface ($\alpha_\mathrm{s} \approx \alpha_{\mathrm{s},0}$) (Pöschl et al., 2007; Shiraiwa et al., 2012). In the absence of condensed-phase reactions, a quasi-static surface accommodation coefficient ($\alpha_\mathrm{ss}$), i.e. the probability for a gas molecule colliding with the surface to enter the quasi-static surface layer, can be calculated as follows (Shiraiwa et al., 2012):

$$\alpha_\mathrm{ss} = \alpha_\mathrm{s} \frac{J_\mathrm{s,ss}}{J_\mathrm{d} + J_\mathrm{s,ss}} = \alpha_\mathrm{s} \frac{k_\mathrm{s,ss}}{k_\mathrm{d} + k_\mathrm{s,ss}} \tag{A1}$$





Here $J_d$ is the desorption flux of Z and $k_d$ is the corresponding first-order rate coefficient; $J_{s,ss}$ and
$k_{s,ss}$ represent the flux and first-order rate coefficient of transfer between the sorption layer and the
quasi-static surface layer. The probability for an individual gas molecule colliding with the surface
to enter the bulk with a penetration depth $x$ can be described by a penetration depth-dependent
mass accommodation coefficient, $\alpha(x)$, defined as follows:
$$\alpha(x) = \alpha_{ss} \frac{\Psi_{ss,bx}}{1 - \Psi_{ss,s}\Psi_{s,ss}} \tag{A2}$$

Here $\Psi_{s,ss}$ is the probability for Z in the sorption layer to enter the quasi-static surface layer and
$\Psi_{ss,bx}$ and $\Psi_{ss,s}$ are the probabilities for Z in the quasi-static surface layer to enter the bulk with the
penetration depth of $x$ or the sorption layer, respectively, which are determined by the
corresponding fluxes and first-order rate coefficients of mass transport (Shiraiwa et al., 2012):
$$\Psi_{s,ss} = J_{s,ss} /(J_{s,ss} + J_{des}) = k_{s,ss} /(k_{s,ss} + k_d) \tag{A3}$$

$$\Psi_{ss,s} = J_{ss,s} /(J_{ss,bx} + J_{ss,s}) = k_{ss,s} /(k_{ss,bx} + k_{ss,s}) \tag{A4}$$

$$\Psi_{ss,bx} = J_{ss,bx} /(J_{ss,bx} + J_{ss,s}) = k_{ss,bx} /(k_{ss,bx} + k_{ss,s}) \tag{A5}$$

Inserting Eqs. (A3)-(A5) in Eq. (A2) leads to:
$$\alpha(x) = \alpha_s \frac{k_{s,ss}}{k_d + k_{s,ss}} \frac{\frac{k_{ss,bx}}{k_{ss,bx} + k_{ss,s}}}{1 - \frac{k_{ss,s}}{k_{ss,bx} + k_{ss,s}} \cdot \frac{k_{s,ss}}{k_{s,ss} + k_d}}$$

$$= \alpha_s \frac{k_{s,ss}k_{ss,bx}}{(k_d + k_{s,ss})(k_{ss,bx} + k_{ss,s})\left(1 - \frac{k_{ss,s}}{k_{ss,bx} + k_{ss,s}} \cdot \frac{k_{s,ss}}{k_{s,ss} + k_d}\right)}$$

$$= \alpha_s \frac{k_{s,ss}k_{ss,bx}}{(k_d + k_{s,ss})(k_{ss,bx} + k_{ss,s}) - k_{ss,s}k_{s,ss}}$$

$$= \alpha_s \frac{k_{s,ss}k_{ss,bx}}{k_d k_{ss,bx} + k_{s,ss}k_{ss,bx} + k_d k_{ss,s}} = \alpha_s \frac{1}{1 + \frac{k_d k_{ss,s} + k_d k_{ss,bx}}{k_{s,ss}k_{ss,bx}}}$$

$$= \alpha_s \frac{1}{1 + \frac{k_d}{k_{s,ss}}\frac{k_{ss,s} + k_{ss,bx}}{k_{ss,bx}}} = \alpha_s \frac{1}{1 + \frac{k_d}{k_{s,ss}}\left(1 + \frac{k_{ss,s}}{k_{ss,bx}}\right)}$$

$$\tag{A6}$$

The first-order rate coefficients of adsorption and desorption are given by $k_a = \alpha_s \, \omega \, / \, 4$ and $k_d =$
$1/\tau_d$, respectively, where $\omega$ (cm s$^{-1}$) is the mean thermal velocity of Z in the gas phase and $\tau_d$ is the
lifetime of desorption from the sorption layer (Pöschl et al., 2007; Shiraiwa et al., 2012). The rate





coefficient of mass transfer between sorption layer and quasi-static surface layer can be estimated
based on the Fick's first law of diffusion considering that a molecule in the sorption layer needs to
travel a distance of $\delta$ to move into the quasi-static surface layer: $k_{ss,s} \approx D_b / \delta^2$ (Shiraiwa et al.,
2012). An estimate for $k_{s,ss}$ can be determined considering mass transport under equilibrium
conditions, where  mass balance implies $J_{s,ss} = J_{ss,s}$, i.e., $k_{s,ss} [Z]_{s,eq} = k_{ss,s} [Z]_{ss,eq}$, and $J_{des} = J_{ads}$, i.e.,
$k_d [Z]_{s,eq} = k_a [Z]_{g,eq}$ (Shiraiwa et al., 2012). Here $[Z]_{g,eq}$, $[Z]_{s,eq}$, and $[Z]_{ss,eq}$ are the equilibrium or
solubility saturation number concentrations of Z in the gas phase, on the sorption layer, and in the
quasi-static surface layer, respectively:
$$k_{s,ss} = k_{ss,s} \frac{k_d [Z]_{ss,eq}}{k_a [Z]_{g,eq}} \quad (A7)$$

$$\frac{k_d}{k_{s,ss}} = \frac{k_a}{k_{ss,s}} \frac{[Z]_{g,eq}}{[Z]_{ss,eq}} = \frac{k_a}{k_{ss,s}} \frac{[Z]_{g,eq}}{[Z]_{b,eq} \delta} \quad (A8)$$


In analogy, the first-order rate coefficient $k_{bx,ss}$ can be estimated based on the Fick's first law of
diffusion, considering that a molecule Z at penetration depth $x$ in the bulk needs to travel a distance
of $x - \delta$ to move into the quasi-static surface layer (Fig. A1): $k_{bx,ss} \approx D_b/(x - \delta)$. Under equilibrium
conditions, $J_{ss,bx} = J_{bx,ss}$ and $k_{ss,bx} [Z]_{ss,eq} = k_{bx,ss} [Z]_{b,eq}$ which leads to $k_{ss,bx} = k_{bx,ss}/\delta = D_b/(\delta(x - \delta))$
assuming ideal mixing conditions and $[Z]_{b,eq} = [Z]_{ss,eq}/\delta$ (Shiraiwa et al., 2012). Thus, $k_{ss,s} / k_{ss,bx} =$
$(D_b / \delta^2) / (D_b / (\delta(x - \delta))) = (x - \delta) / \delta$.

Based on the absorptive partitioning theory (Donahue et al., 2006; Pankow, 1994),


$$C^0 = \frac{C^g}{C^P} C_{OA} \quad (A9)$$

where $C^0$ ($\mu$g m$^{-3}$) is the pure compound saturation mass concentration, $C^g$ and $C^P$ ($\mu$g m$^{-3}$) are the
gas-phase and particle-phase mass concentrations of the compound Z, respectively, and $C_{OA}$ ($\mu$g
m$^{-3}$) is the total organic aerosol mass concentration. $C^g$ and $[Z]_{g,eq}$ are related through the following
equation:
$$C^g = \frac{[Z]_{g,eq} M}{N_A} \cdot 10^{12} \frac{\mu g}{g} \frac{m^{-3}}{cm^{-3}} \quad (A10)$$

where $M$ is the molar mass of compound Z. $[Z]_{g,eq}$ is the equilibrium (saturation) number
concentration of Z in the gas phase. $[Z]_{g,eq}$ can be calculated using the saturation vapor pressure $p$:


$[Z]_{g,eq} = p\,N_A\,/\,(R\,T)$ where $N_A$ is the Avogadro number, $R$ is the gas constant, and $T$ is the
temperature. $[Z]_{b,eq}$ corresponds to the ratio between the number concentration of Z in the particle
phase (per $m^3$ of air) to the particle volume concentration ($m^3$ per $m^3$ of air), which can be
expressed using $C_Z^{PM}$ and $C_{OA}$ with the particle density $\rho_P$ (g cm$^{-3}$):
$$[Z]_{b,eq} = \frac{\dfrac{C^P}{M} N_A}{\dfrac{C_{OA}}{\rho}} = \frac{C^P N_A \rho_P}{C_{OA} M} \quad (A11)$$

Combining Eq. (A9) – (A11) would lead to:
$$\frac{[Z]_{g,eq}}{[Z]_{b,eq}} = \frac{\dfrac{C^g\,N_A}{M}}{\dfrac{C^P\,N_A\rho_P}{C_{OA}M}} \cdot 10^{-12} = \frac{C^g}{C^P}\,C_{OA}\frac{1}{\rho_P} \cdot 10^{-12} = \frac{C^0}{\rho_P} \cdot 10^{-12}\,\frac{\text{g}}{\mu\text{g}}\frac{\text{cm}^{-3}}{\text{m}^{-3}} \quad (A12)$$

Inserting Eq. (A8) into Eq. (A6) and combination with Eq. (A12) leads to:
$$\alpha(x) = \alpha_s \frac{1}{1 + \dfrac{k_a}{k_{ss,s}}\dfrac{[Z]_{g,eq}}{[Z]_{b,eq}}\delta\left(1 + \dfrac{x-\delta}{\delta}\right)} = \alpha_s \frac{1}{1 + \dfrac{\alpha_s \omega C^0}{4\,D_b\,\rho_P}x \cdot 10^{-12}\,\dfrac{\text{g}}{\mu\text{g}}\dfrac{\text{cm}^{-3}}{\text{m}^{-3}}} \quad (A13)$$


**Acknowledgements.** MS acknowledges funding by the National Science Foundation (AGS-
1654104) and the Department of Energy (DE-SC0018349). We thank Jose Jimenez (CU Boulder)
for stimulating discussions and for sharing published data and experimental information as
presented in Figure 4.

**Author contributions.** MS and UP designed the study, analyzed the data, and wrote the paper.
MS conducted kinetic modeling.

**Competing interests**. The authors declare that they have no conflict of interest.

**Data availability.** The simulation data may be obtained from the corresponding author upon
request.

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
