# Peer review of "Mass Accommodation and Gas-Particle Partitioning in Secondary Organic Aerosols: Dependence on Diffusivity, Volatility, Particle-phase Reactions, and Penetration Depth Manabu Shiraiwa1,\* and Ulrich Pöschl2,\* 1. Department of Chemistry, University of California, Irvine, CA92625"

_Atmospheric Chemistry and Physics, 2020_

## Referee Comment (RC1) · Anonymous Referee #3 · 11 Nov 2020

To date, Fuchs-Sutugin (F-S) approximation of mass-transport kinetics at the gas-particle interface with a fixed mass accommodation coefficient has been commonly in many models. In this work, the authors introduced an effective mass accommodation coefficient which considers penetration depth, surface accommodation coefficient, volatility, bulk diffusivity, and particle-phase reaction rate constant. They also suggest that this new coefficient and its dependence on these various parameters should be considered in the future when the F-S approximation is used to simulate gas-particle

interactions, in particular for viscous or semi-solid organic aerosols, which are commonly found in the atmosphere. The paper is well written and concise. It also provide new method for simulating SOA formation and evaluation. I support the publication of this work in ACP and have some minor comments below.

Comments

Line 194, "We simulate the temporal evolution of partitioning and equilibration of semi-volatile organic compounds (SVOC) with C0 = 100 $\mu$g m-3 and Dg = 0.1 cm2 s-1 interacting with non-volatile seed particles with a number concentration of 5000 cm-3, an initial diameter of 200 nm, and a surface accommodation coefficient as = a(0) = 1. For the SVOC, we assume initial gas- and particle-phase concentrations of 2 $\mu$g m-3 and 0 $\mu$g m-3, respectively. The particles are assumed to be either liquid with a bulk diffusion coefficient Db = 10-7 cm2 s-1 or semisolid with Db = 10-15 cm2 s-1." Can the authors elaborate and justify why these parameters are chosen for their simulations? A typical condition for ambient conditions or laboratory studies?

Line 214, "For liquid particles with fast surface-bulk exchange and bulk diffusion (Db = 10-7 cm2 s-1), a(x) remains close to as = a(0) = 1, and all models yield the same result of fast mass transfer from the gas to particle phase and equilibration within one second." Have the authors shown these results in the manuscript?

Line 235, "At long timescales (> 1 h), the partitioning is reasonable well captured by both the MOSAIC approximation using a two-film approach of bulk diffusion (Zaveri et al., 2014) as well as the simple F-S approximation accounting for the influence of penetration depth with the effective mass accommodation coefficient, aeff, newly introduced this study." Can the authors comment how the simple F-S approximation accounting for the influence of penetration depth with the effective mass accommodation coefficient can be improved or used at short timescales (< 1min)?

Line 253, "Model results for SVOC partitioning plus reactive uptake with different rate coefficients in semisolid aerosol particles are shown in Figure 2b." Can the authors

elaborate how to choose these first-order bulk reaction rate coefficients ($k_b$ = 0, 10-4, 10-3, 0.01, 0.1 s-1)?

Line 263, "Figures 3a and 3b show how the effective mass accommodation coefficient $a_{eff}$ depends on volatility and bulk diffusivity as related to particle phase state and viscosity according to the Stokes-Einstein relation (Shiraiwa et al., 2011)." What are the timescale used in these simulations (e.g. < 1min, 1min to 1hr or > 1hr)? Would the simulated results affect by the timescale selected for the simulations (e.g. < 1min vs. > 1hr)?

Line 263, For Figure 4, what are the timescale used in these simulations (e.g. < 1min, 1min to 1hr or > 1hr)? Would the simulated results affect by the timescale chosen for the simulations (e.g. < 1min vs. > 1hr)?

Summary and conclusions. In this section, can the authors discuss how an effective mass accommodation coefficient can be extended to apply for aerosols containing both inorganic and organic species? How the phase separation and morphology can be accounted in the effective mass accommodation coefficient?

---

## Referee Comment (RC2) · Anonymous Referee #4 · 15 Dec 2020

This manuscript introduces a quantity termed as "effective mass accommodation coefficient" which essentially accounts for particulate phase transport and reactions unlike the original mass accommodation coefficient introduced by Fuchs. The authors argue that such a quantity will be useful for e.g. large-scale modeling applications aiming to understand secondary organic aerosol (SOA) instead of having to conduct explicit calculations resolving the particulate phase and the gas-particle interface at different conditions. This manuscript is a welcome addition to the discussion on the kinetics related

to SOA formation and growth, and fits well within the scope of ACP. I appreciate the clear distinction between the effective mass accommodation coefficient defined here and the "original" mass accommodation coefficient, as these concepts have unfortunately been often confused in recent literature dealing with SOA kinetics. I recommend publication in ACP after the following issues have been adequately addressed:

1. My main comment is related to whether it is appropriate to term the new coefficient an accommodation coefficient to correctly represent the targeted phenomena (i.e. the particle phase transport and chemistry at the interface region). According to the original Fuchs-Sutugin formulation (Eq. 2 in the manuscript) the accommodation coefficient is a quantity relevant in the kinetic regime and its impact on the mass flux towards the particle disappears at the limit of small Knudsen numbers (i.e. for large particles if pressure is assumed constant). Will the presented formulation of the effective mass accommodation coefficient give the correct dependence on the particle size? Is it physically correct that the effect of the effective accommodation coefficient also disappears at the limit of Kn -> 0? The authors should elaborate on this and justify their choice of representing the particle phase phenomena as an effective accommodation coefficient instead of a flux correction factor.

2. The authors end the abstract with a rather strong statement: "Our findings challenge the approach of traditional SOA models using the Fuchs-Sutugin approximation of mass transfer kinetics with a fixed mass accommodation coefficient regardless of particle phase state and penetration depth. The effective mass accommodation coefficient introduced in this study provides an efficient new way of accounting for the influence of volatility, diffusivity, and particle-phase reactions on SOA partitioning in process models as well as in regional and global air quality models." At the same time, the authors also acknowledge the fact that the particle-phase transport is only relevant at rather low RHs. While it is true that at some conditions (like in the free troposphere) the semi-solid state of the SOA is highly relevant, I would suggest softening the statements related to the implications of this study for global and regional SOA modeling

when it comes to conditions representative of surface-level RH and temperature.

3. Lines 94-97: The authors state: "Molecular dynamics simulations (Julin et al., 2014; Von Domaros et al., 2020) and a recent SOA chamber study (Liu et al., 2019) suggest that the mass accommodation coefficients for semi-volatile organic molecules on organic substrates are close to unity. Measurement-derived mass accommodation coefficients reported from thermodenuder investigations of SOA volatility distributions, however, were one to three orders of magnitude lower (Kostenidou et al., 2018; Lee et al., 2010; Saleh et al., 2011)." I think it should be noted that e.g. the studies by Lee et al. and Saleh et al. have been subject to a relatively high uncertainty in the assumed saturation concentrations of the studied species (e.g. at the time of these studies the auto-oxidation reactions generating ELVOCs in SOA mixtures were not established like they are today). Therefore, I think these experimental studies studying complex SOA mixtures are hardly comparable to the more recent MD simulations and laboratory studies. Please revise to present a more relevant comparison.

4. The presented effective mass accommodation coefficient is dependent on a variable called the "penetration depth". How should this parameter be defined for ambient SOA mixtures? This is rather important for the usefulness of the proposed approach and further elaboration on this would be important in the discussion of the results.

---

## Author Comment (AC1) · 19 Dec 2020

**Response to Referee comments** (comments in black, response in blue)

**Anonymous Referee #3**

To date, Fuchs-Sutugin (F-S) approximation of mass-transport kinetics at the gas-particle interface with a fixed mass accommodation coefficient has been commonly in many models. In this work, the authors introduced an effective mass accommodation coefficient which considers penetration depth, surface accommodation coefficient, volatility, bulk diffusivity, and particle-phase reaction rate constant. They also suggest that this new coefficient and its dependence on these various parameters should be considered in the future when the F-S approximation is used to simulate gas-particle C1 interactions, in particular for viscous or semi-solid organic aerosols, which are commonly found in the atmosphere. The paper is well written and concise. It also provide new method for simulating SOA formation and evaluation. I support the publication of this work in ACP and have some minor comments below.

We thank this Referee for the review and positive evaluation of our manuscript.

Comments Line 194, "We simulate the temporal evolution of partitioning and equilibration of semivolatile organic compounds (SVOC) with $C_0$ = 100 µg m$^{-3}$ and $D_g$ = 0.1 cm$^2$ s$^{-1}$ interacting with non-volatile seed particles with a number concentration of 5000 cm-3, an initial diameter of 200 nm, and a surface accommodation coefficient as = a(0) = 1. For the SVOC, we assume initial gas- and particle-phase concentrations of 2 µg m$^{-3}$ and 0 µg m$^{-3}$, respectively. The particles are assumed to be either liquid with a bulk diffusion coefficient $D_b$ = 10$^{-7}$ cm$^2$ s$^{-1}$ or semisolid with $D_b$ = 10$^{-15}$ cm$^2$ s$^{-1}$." Can the authors elaborate and justify why these parameters are chosen for their simulations? A typical condition for ambient conditions or laboratory studies?

We chose these values as they are typical values of SVOC volatility and viscosity for SOA based on previous measurements. In addition, these values were used in Zaveri et al. (2014) and it is easy to refer and compare with this study. We clarify this point in the revised manuscript.

Line 214, "For liquid particles with fast surface-bulk exchange and bulk diffusion ($D_b$ = 10$^{-7}$ cm$^2$ s$^{-1}$), a(x) remains close to as = a(0) = 1, and all models yield the same result of fast mass transfer from the gas to particle phase and equilibration within one second." Have the authors shown these results in the manuscript?

For simplicity and legibility of Fig. 1a, we do not show these results in the manuscript, but all model lines obtained for this scenario are overlapping as shown below. We clarify this point in the revised manuscript.

[Figure]

Line 235, "At long timescales (> 1 h), the partitioning is reasonable well captured by both the MOSAIC approximation using a two-film approach of bulk diffusion (Zaveri et al., 2014) as well as the simple F-S approximation accounting for the influence of penetration depth with the effective mass accommodation coefficient, aeff, newly introduced this study." Can the authors comment how the simple F-S approximation accounting for the influence of penetration depth with the effective mass accommodation coefficient can be improved or used at short timescales (< 1min)?

The F-S approach with $\alpha_{eff}$ underestimates partitioning at short timescales because the particle phase does not reach a quasi-steady state and corresponding bulk concentration gradient, whereas the application of $\alpha_{eff}$ is based on the assumption of an effective penetration depth of $r_p/5$ (Eq. 6). This is an inherent limitation for both the F-S approximation and the $\alpha_{eff}$ approach which are assuming a quasi-steady state. The time to reach a quasi-steady state depends on bulk diffusivity, particle radius, and particle-phase reaction rate coefficient (e.g., Fig. 5 in Zaveri et al., 2014). At shorter timescales, we recommend the use of kinetic multilayer models or similarly detailed modeling approaches that can resolve transient conditions. We clarify this point in the revised manuscript.

Line 253, "Model results for SVOC partitioning plus reactive uptake with different rate coefficients in semisolid aerosol particles are shown in Figure 2b." Can the authors C2 elaborate how to choose these first-order bulk reaction rate coefficients (kb = 0, 10-4, 10-3, 0.01, 0.1 s-1)?

The $k_b$ value would vary for different compounds. A study has shown that chemical half-lives of highly oxygenated organic molecules are shorter than one hour (Krapf et al., 2016), corresponding to $k_b > \sim 2 \times 10^{-4}$. First-order decomposition rate coefficients for organic hydroperoxides in SOA were reported in the range of $10^{-6} - 1.5 \times 10^{-3}$ (Tong et al., 2016; Tong et al., 2018; Wei et al., 2020) and can be enhanced by photolysis (Badali et al., 2015; Epstein et al., 2014) or Fenton-like reactions in the presence of transition metal ions (Goldstein and Meyerstein, 1999). We add this aspect in the revised manuscript.

Line 263, "Figures 3a and 3b show how the effective mass accommodation coefficient aeff depends on volatility and bulk diffusivity as related to particle phase state and viscosity according to the Stokes-Einstein relation (Shiraiwa et al., 2011)." What are the timescale used in these simulations (e.g. < 1min, 1min to 1hr or > 1hr)? Would the simulated results affect by the timescale selected for the simulations (e.g. < 1min vs. > 1hr)?
Line 263, For Figure 4, what are the timescale used in these simulations (e.g. < 1min, 1min to 1hr or > 1hr)? Would the simulated results affect by the timescale chosen for the simulations (e.g. < 1min vs. > 1hr)?

The results presented in Figures 3 and 4 were not obtained by numerical simulations, but were calculated with the analytical equations Eq. 5 & 6 under the assumption of quasi-steady-state conditions. We clarify this point in the revised manuscript.

Summary and conclusions. In this section, can the authors discuss how an effective mass accommodation coefficient can be extended to apply for aerosols containing both inorganic and

organic species? How the phase separation and morphology can be accounted in the effective mass accommodation coefficient?

The penetration depth and related formulations presented in this study assume that organic particles (which can be mixed with inorganic components) are homogenous without considering potential gradients of bulk diffusivity. As pointed out, mixed organic-inorganic particles often undergo liquid-liquid phase separation. Additional work is necessary to develop advanced formulations for mass transfer of gas-phase species to particles with complex morphology. This aspect goes beyond the scope of this current study and may require further studies as stated in the last sentence of the manuscript:

"mixed organic-inorganic particles often undergo liquid-liquid phase separation at moderate and high RH (Krieger et al., 2012; You et al., 2014; Zuend and Seinfeld, 2012), and liquid-liquid phase separation can also occur for purely organic particles (Song et al., 2017). The interplay of particle phase state and phase separation can further impact SOA partitioning (Shiraiwa et al., 2013b). In such complex particle morphologies with multiple phases, gradients and discontinuities of diffusivity may occur within the particle bulk and require more advanced modeling approaches of gas-particle interaction kinetics to be addressed in future studies."

---

## Author Comment (AC2) · 19 Dec 2020

This manuscript introduces a quantity termed as "effective mass accommodation coefficient" which essentially accounts for particulate phase transport and reactions unlike the original mass accommodation coefficient introduced by Fuchs. The authors argue that such a quantity will be useful for e.g. large-scale modeling applications aiming to understand secondary organic aerosol (SOA) instead of having to conduct explicit calculations resolving the particulate phase and the gas-particle interface at different conditions. This manuscript is a welcome addition to the discussion on the kinetics related to SOA formation and growth, and fits well within the scope of ACP. I appreciate the clear distinction between the effective mass accommodation coefficient defined here and the "original" mass accommodation coefficient, as these concepts have unfortunately been often confused in recent literature dealing with SOA kinetics. I recommend publication in ACP after the following issues have been adequately addressed:

We thank this Referee for the review and positive evaluation of our manuscript.

1. My main comment is related to whether it is appropriate to term the new coefficient an accommodation coefficient to correctly represent the targeted phenomena (i.e. the particle phase transport and chemistry at the interface region). According to the original Fuchs-Sutugin formulation (Eq. 2 in the manuscript) the accommodation coefficient is a quantity relevant in the kinetic regime and its impact on the mass flux towards the particle disappears at the limit of small Knudsen numbers (i.e. for large particles if pressure is assumed constant). Will the presented formulation of the effective mass accommodation coefficient give the correct dependence on the particle size? Is it physically correct that the effect of the effective accommodation coefficient also disappears at the limit of Kn -> 0? The authors should elaborate on this and justify their choice of representing the particle phase phenomena as an effective accommodation coefficient instead of a flux correction factor.

The Fuchs-Sutugin approach for the transition regime is validated by experiments with ~$6\times10^{-3}$ < Kn < 3 (Fig. 12.3 in Seinfeld & Pandis, 2016). With a typical mean free path of ~100 nm for SOA compounds, it corresponds to particle radius of ~33 nm < $r_p$ < ~17 μm, covering a typical size range of SOA particles observed in ambient atmosphere and laboratory experiments. For larger particles with Kn → 0, $\beta \to 0.75\alpha / 0.75\alpha = 1$ and the mass transfer flux converges to a solution for the continuum regime. As pointed out, the influence of mass accommodation on the F-S gas diffusion flux becomes negligibly small in the continuum regime where mass transfer is limited by gas-phase diffusion driven by concentration gradients between the gas phase and the particle surface. In fact, we consider the surface accommodation coefficient as a fundamental kinetic parameter as defined by PRA (2007) – regardless of the specific mass transfer regime – and not just as a parameter defined by Eq. 2. The effective accommodation coefficient, on the other hand, comprises both the fundamental quantity $\alpha_s$ and a flux correction depending on the effective penetration depth as defined by Eqs. (5)-(10). We clarify this point in the revised manuscript (at the end of the Theory and Methods section).

2. The authors end the abstract with a rather strong statement: "Our findings challenge the approach of traditional SOA models using the Fuchs-Sutugin approximation of mass transfer kinetics with a fixed mass accommodation coefficient regardless of particle phase state and penetration depth.

The effective mass accommodation coefficient introduced in this study provides an efficient new way of accounting for the influence of volatility, diffusivity, and particle-phase reactions on SOA partitioning in process models as well as in regional and global air quality models." At the same time, the authors also acknowledge the fact that the particle-phase transport is only relevant at rather low RHs. While it is true that at some conditions (like in the free troposphere) the semi-solid state of the SOA is highly relevant, I would suggest softening the statements related to the implications of this study for global and regional SOA modeling when it comes to conditions representative of surface-level RH and temperature.

We appreciate the suggestion and explain in both the revised abstract and conclusions that kinetic limitations of bulk diffusion may not be critical for partitioning into liquid SOA particles in the planetary boundary layer at high T and RH but likely important for semi-solid or glassy SOA at low RH and T.

Beyond that, we would like to emphasize that effective mass accommodation coefficient is a very efficient method to properly treat gas-particle partitioning in large-scale models, because it is easily applicable for liquid, semi-solid, and solid particles as function of standard physicochemical parameters. While the kinetic limitation of bulk diffusion is relevant only for viscous and solid particles leading to reduced $\alpha_{\text{eff}}$ (e.g., Figure 3), the same formulations (Eq. 1-10) are applicable for particles with different phase states, because Eq. 5 explicitly accounts for bulk diffusivity.

3. Lines 94-97: The authors state: "Molecular dynamics simulations (Julin et al., 2014; Von Domaros et al., 2020) and a recent SOA chamber study (Liu et al., 2019) suggest that the mass accommodation coefficients for semi-volatile organic molecules on organic substrates are close to unity. Measurement-derived mass accommodation coefficients reported from thermodenuder investigations of SOA volatility distributions, however, were one to three orders of magnitude lower (Kostenidou et al., 2018; Lee et al., 2010; Saleh et al., 2011)." I think it should be noted that e.g. the studies by Lee et al. and Saleh et al. have been subject to a relatively high uncertainty in the assumed saturation concentrations of the studied species (e.g. at the time of these studies the auto-oxidation reactions generating ELVOCs in SOA mixtures were not established like they are today). Therefore, I think these experimental studies studying complex SOA mixtures are hardly comparable to the more recent MD simulations and laboratory studies. Please revise to present a more relevant comparison.

Saleh et al., 2011 estimated $\alpha$ for dicarboxylic acids with known vapor pressures. Lee et al, 2010 estimated volatility of ambient organic aerosols in Finokalia, Greece, based on thermodenuder measurements with the lowest volatility bin of 0.01 µg m$^{-3}$; as we do not have information on ELVOC/HOM contributions to organic aerosols in Finokalia, it is hard to judge how the luck of knowledge of HOM/ELVOC would have affected their study. As both studies were peer-reviewed and published, we are not in the position to judge/discredit their studies and would like to keep these references to provide a balanced view to readers.

4. The presented effective mass accommodation coefficient is dependent on a variable called the "penetration depth". How should this parameter be defined for ambient SOA mixtures? This is rather important for the usefulness of the proposed approach and further elaboration on this would be important in the discussion of the results.

The penetration depth and related formulations presented in this study assume that particles are homogenous without considering potential gradients of bulk diffusivity. In real ambient SOA complex mixtures, particles may adopt layered structures such as surface crusts (solid/viscous surface layers). Mixed organic-inorganic particles often undergo liquid-liquid phase separation, which can also occur for purely organic particles. For these cases, the penetration depth may be confined to particle shells, which can be smaller than the depth calculated from particle radius with Eq. 6 & 7. To the best of our knowledge, there are few large-scale atmospheric models which resolve kinetic partitioning into inhomogeneous particles. Currently, partitioning into these inhomogeneous particles with diffusivity gradients would need to be treated with a detailed model like KM-GAP, which should be applied to investigate and quantify such impacts on SOA partitioning kinetics. As such, the development of simple parameterizations appears highly challenging, goes well beyond the scope of current study, and may require new experimental techniques and extensive data sets.